# Sleep and motor learning in stroke (SMiLES): a longitudinal study investigating sleep-dependent consolidation of motor sequence learning in the context of recovery after stroke

Matthew Weightman ,[1] Barbara Robinson ,[1] Morgan P Mitchell,[1] Emma Garratt,[2] Rachel Teal,[3] Andrew Rudgewick-Brown,[1] Nele Demeyere,[4] Melanie K Fleming ,[1] Heidi Johansen-Berg[1]

MKF and HJ-B contributed equally.

For numbered affiliations see end of article.

**Correspondence to**
Dr Matthew Weightman;
matthew.weightman@ndcn.ox.ac.uk

## ABSTRACT

**Introduction** There is growing evidence that sleep is disrupted after stroke, with worse sleep relating to poorer motor outcomes. It is also widely acknowledged that consolidation of motor learning, a critical component of poststroke recovery, is sleep-dependent. However, whether the relationship between disrupted sleep and poor outcomes after stroke is related to direct interference of sleep-dependent motor consolidation processes, is currently unknown. Therefore, the aim of the present study is to understand whether measures of motor consolidation mediate the relationship between sleep and clinical motor outcomes post stroke.

**Methods and analysis** We will conduct a longitudinal observational study of up to 150 participants diagnosed with stroke affecting the upper limb. Participants will be recruited and assessed within 7 days of their stroke and followed up at approximately 1 and 6 months. The primary objective of the study is to determine whether sleep in the subacute phase of recovery explains the variability in upper limb motor outcomes after stroke (over and above predicted recovery potential from the Predict Recovery Potential algorithm) and whether this relationship is dependent on consolidation of motor learning. We will also test whether motor consolidation mediates the relationship between sleep and whole-body clinical motor outcomes, whether motor consolidation is associated with specific electrophysiological sleep signals and sleep alterations during subacute recovery.

**Ethics and dissemination** This trial has received both Health Research Authority, Health and Care Research Wales and National Research Ethics Service approval (IRAS: 304135; REC: 22/LO/0353). The results of this trial will help to enhance our understanding of the role of sleep in recovery of motor function after stroke and will be disseminated via presentations at scientific conferences, peer-reviewed publication, public engagement events, stakeholder organisations and other forms of media where appropriate.

---

### STRENGTHS AND LIMITATIONS OF THIS STUDY

⇒ Longitudinal study, with early assessment (≤7 days post stroke) and follow-ups throughout subacute recovery.
⇒ All assessments carried out where the patient is currently residing to reduce burden.
⇒ Inclusion of patients without capacity to consent to increase the diversity of study sample.
⇒ Some patients may not be able to complete all aspects of the study due to level of impairment.
⇒ Likely between-subject variability in the amount of upper extremity therapy/treatment received during study enrolment.

---

**Trial registration number** ClinicalTrials.gov: NCT05746260, registered on 27 February 2023.

## INTRODUCTION

Stroke is a leading cause of worldwide disability, with over 50% of stroke survivors experiencing long-term motor impairment.[1–3] It is widely believed that good recovery of movement after stroke is dependent on motor learning.[4] Indeed, most current rehabilitative practises aiming to improve motor function after stroke place a key focus on learning new, or relearning old, ways in which to successfully carry out activities of daily living.[5 6] Motor learning is thought to be especially important for poststroke recovery due to its capacity to drive structural and functional changes in the brain, termed neuroplasticity.[7 8] As good rehabilitation outcomes after stroke are dependent on the ability of the brain to learn and reorganise, it is critical

we understand how neuroplasticity occurs and how it can be influenced in the context of recovery.

A key element of our understanding of neuroplasticity—which has yet to be fully explored in the context of stroke rehabilitation—is consolidation of memories during sleep.[4] Improvements in motor learning are known to not only occur online (during practise itself) but also offline, during periods of rest.[9] For some time now, sleep has been proposed to be actively engaged in the offline consolidation of motor memories.[10–12] The reactivation of task-related neural activity during sleep is believed to underpin offline memory processing, by transforming labile memory traces into persistent representations, readily available for retrieval.[13 14] These reactivations are also closely linked to fine-tuned interactions between electrophysiological signatures of slow wave sleep, namely; neocortical slow oscillations and thalamocortical sleep spindles, known to be key correlates of memory consolidation.[15–18] Taken together, this evidence highlights the importance of sleep to the offline consolidation of motor memories.

Notwithstanding, sleep disruption is commonplace after stroke, with more fragmented sleep, reduced sleep efficiency and less total sleep time commonly reported.[19–22] Evidence suggests that poor sleep after brain injury, including stroke, is often associated with worse motor/cognitive impairments, quality of life and mood.[22–24] Furthermore, we and others have demonstrated that sleep disruption during rehabilitation after stroke and/or brain injury is associated with slower functional recovery and worse motor outcomes.[23 25] However, whether this relationship exists because impaired sleep post stroke directly disrupts memory consolidation processes, or reflects other factors that influence sleep and clinical outcomes, for example, depression, motivation etc, is yet unknown. Given that clinical gains during therapy are likely to depend on both improvements during training and sleep, it is reasonable to suggest that poor motor outcomes may be, at least in part, due to disrupted sleep and thereby associated memory processes.

### Trial objectives

Considering the link between sleep disruption and recovery, the aim of the present study is to understand to what extent behavioural measures of motor consolidation mediate the relationship between sleep quality and clinical motor outcomes post stroke. We will investigate whether sleep measures in the subacute phase explain variation in motor outcomes (over and above variation explained by baseline assessment, and other covariates) and whether this relationship is mediated by variation in behavioural measures of overnight consolidation. More specifically, we aim to:

Primary objectives:
1. Test whether sleep measures at ~1 month post stroke explain variability in recovery of movement of the paretic upper limb at ~6 months.

2. Assess whether the relationship between sleep and recovery after stroke depends on motor consolidation.

Secondary objective:
1. Determine if motor consolidation at ~1 month post stroke mediates the relationship between sleep and clinical outcomes, such as whole-body motor impairment, mobility and hand function of the more-affected side, measured at ~6 months post stroke.

Exploratory objectives:
1. Explore whether motor consolidation is associated with specific electrophysiological markers of sleep after stroke.
2. Investigate changes in sleep alongside recovery over the first 6 months after stroke.
3. Explore correlations between sleep disruption and motor function between ~1 and 6 months post stroke.

## METHODS AND ANALYSIS
### Participant recruitment

Up to 150 participants, diagnosed with stroke affecting the upper limb, will be recruited. All participants will be identified from stroke wards at the acute stage post stroke (≤7 days). Patients will be approached by a member of the clinical team or a clinical research facilitator (eg, research nurse) to determine potential interest and provide information about the study. On verbal consent (or in case of expressive speech problems, a clear non-verbal indication, eg, a nod), a researcher will approach interested patients to go through the informed consent process for the study. It will be made clear that participation is voluntary and will not affect their medical or rehabilitative care. Where the mental capacity to provide informed consent is in doubt, a mental capacity assessment for consenting to the research study will be conducted and appropriate consultee advice will be sought from a familial carer, other relative or close friend if appropriate (figure 1). Inclusion and exclusion criteria are shown below. The trial was registered on 27 February 2023 (prior to recruitment of the first participant; NCT05746260), opened to recruitment in March 2023, with a planned trial end date of March 2027.

### Sample size calculations

Based on findings from previous work in inpatients with brain injury,[23] we predict that sleep measures increase variance in motor outcomes explained by ~10%. Therefore, to achieve a power of 0.9 (1−β error probability) with a significance level (alpha) of <0.05, a total sample size of ~n=100 is recommended (calculated using G*Power; linear multiple regression, with a fixed model). In anticipation of participant attrition, we will aim to recruit 150 participants.

### Inclusion criteria
► Aged 18 years or above.
► Stroke affecting the upper limb, as confirmed by clinical diagnosis (no specific requirement regarding affected side and limb dominance).

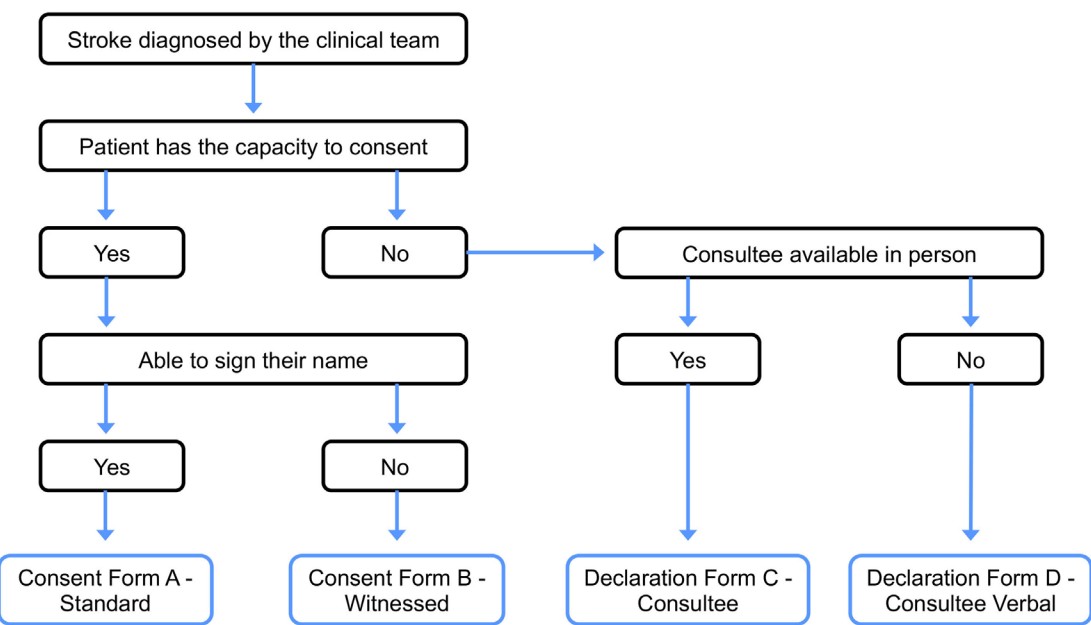

**Figure 1** Consent process, including capacity to consent decision framework.

► The participant is willing and able to give informed consent for participation in the study or a positive opinion from a consultee is provided by a family member or carer (relative or friend) willing to provide personal consultee advice.
► Recruited ≤7 days of stroke.

### Exclusion criteria

► Diagnosis of other neurological condition affecting movement (such as Parkinson's disease, multiple sclerosis, etc).
► Additional exclusion criteria apply to the application of transcranial magnetic stimulation (TMS), which may be required as part of the baseline assessment (Predict Recovery Potential (PREP2) algorithm). Not all participants will require TMS (see Trial design), and ineligibility will not affect participation.
  – Exclusion criteria for TMS procedure include: epilepsy/family history of epilepsy, seizures, metallic implants (above the shoulders), implanted electronics (eg, pacemaker, cochlear implant, medical pump, etc), history of skull fracture or traumatic head injury, history of brain surgery, pregnancy, history of recurring headaches.

### Trial design

The trial will be conducted as a longitudinal observational study. The direct clinical care team or research nurse will identify and screen potential participants, checking their medical history to confirm eligibility. Should participants be interested and meet all eligibility criteria, a researcher will then arrange a consultation for the baseline visit (≤ 7 days post stroke), where they will go over the details of the study, answer any remaining questions and complete the informed consent process (figure 1). Baseline assessment will be conducted as soon as possible thereafter.

Following consent, participants will be enrolled in the trial and assigned a unique study code, and the direct care team will be informed (verbally) that the patient has consented to take part in the study. We will also inform each participant's general practitioner (GP) of consent in the study.

### Outcome measures

Our main outcomes will be used to determine whether sleep in the subacute phase of recovery after stroke explains the variability seen in upper limb motor outcomes and whether this relationship is dependent on overnight consolidation of motor memories. The main outcome measures are described in detail in the descriptions of assessment sessions below. Briefly:

*Sleep disruption* will be measured via 7-night actigraphy recordings and sleep diaries, indicating estimated total sleep time, wakefulness after sleep onset (WASO) and sleep fragmentation.

*Symptoms of insomnia* will be recorded using the Sleep Condition Indicator Questionnaire (SCI),[26 27] with higher scores indicating fewer symptoms of insomnia (range 0–32).

*Motor consolidation* will be measured at approximately 1 month after stroke and will describe the change in performance (response time/accuracy) from evening training to morning retest in the adapted Serial Reaction Time Task (SRTT).[12 28]

*Upper limb function* will be assessed at approximately 6 months post stroke using the Action Research Arm Test (ARAT).[29] Scores range between 0 and 57, with higher scores indicates better upper limb ability.

Secondary outcome measures include: whole-body motor impairment, hand dexterity and mobility at ~6 months after stroke, assessed via the Fugl-Meyer

Assessment (FMA),[30] Nine Hole Peg Test (9HPT)[31] and Rivermead Mobility Index (RMI),[32] respectively (higher scores across all measurements indicates better movement/performance). These outcomes will help to determine whether motor consolidation mediates the relationship between sleep and different aspects of motor recovery. Actigraphy derived sleep measures and SCI scores at 1 and 6 months post stroke will be used to explore changes in sleep quality during subacute recovery. Additionally, electroencephalogram (EEG)-derived sleep measures (eg, slow oscillation amplitude, sleep spindle density, slow oscillation-sleep spindle coupling, etc) collected in a subset of participants at 1 month post stroke will be used to discern whether consolidation of motor learning is associated with specific sleep signals in this population.

Other prespecified measures include: symptoms of depression via the Patient Health Questionnaire (PHQ-8),[33] stroke severity/disability via the modified Ranking Scale (mRS) and physical activity (actigraphy recordings) assessed at approximately 1 and 6 months post stroke. In addition to the risk of obstructive sleep apnoea using the Obstructive Sleep Apnoea Questionnaire (STOP-Bang Questionnaire)[34] and presence of sleep disorders[35] at 1 month post stroke. These will be used as covariates in analyses as appropriate.

## Baseline assessment

The baseline assessment will take place ≤7 days post stroke. We will record participant demographics, including: age, sex, ethnicity, weight, height, stroke type, stroke date and current medications (figure 2).

To characterise expected upper limb recovery potential, we will follow procedures for the PREP2 algorithm.[36] In the initial days post stroke, the PREP2 algorithm combines neurological biomarkers and clinical measures, to predict upper limb functional outcomes at 3 months (figure 3). Patients can be categorised into one of four different recovery categories: excellent, good, poor and limited, with predictions shown to be accurate for up to 80% of patients.[36 37] A Shoulder Abduction and Finger Extension (SAFE) Score will be calculated using the Medical Research Council grades to determine participants' strength in each of these movements. Movements are scored between 0 and 5, where 0 represents no palpable muscle activity and 5 describes normal strength and range of movement. Participants with an SAFE Score of <5 will require TMS to establish the presence of motor evoked potentials (MEPs) in the paretic upper limb for further categorisation. Participants will be screened separately for TMS safety and participants can still take part in the study if unable to undergo TMS. If a participant requires TMS—and is deemed appropriate to do so as per

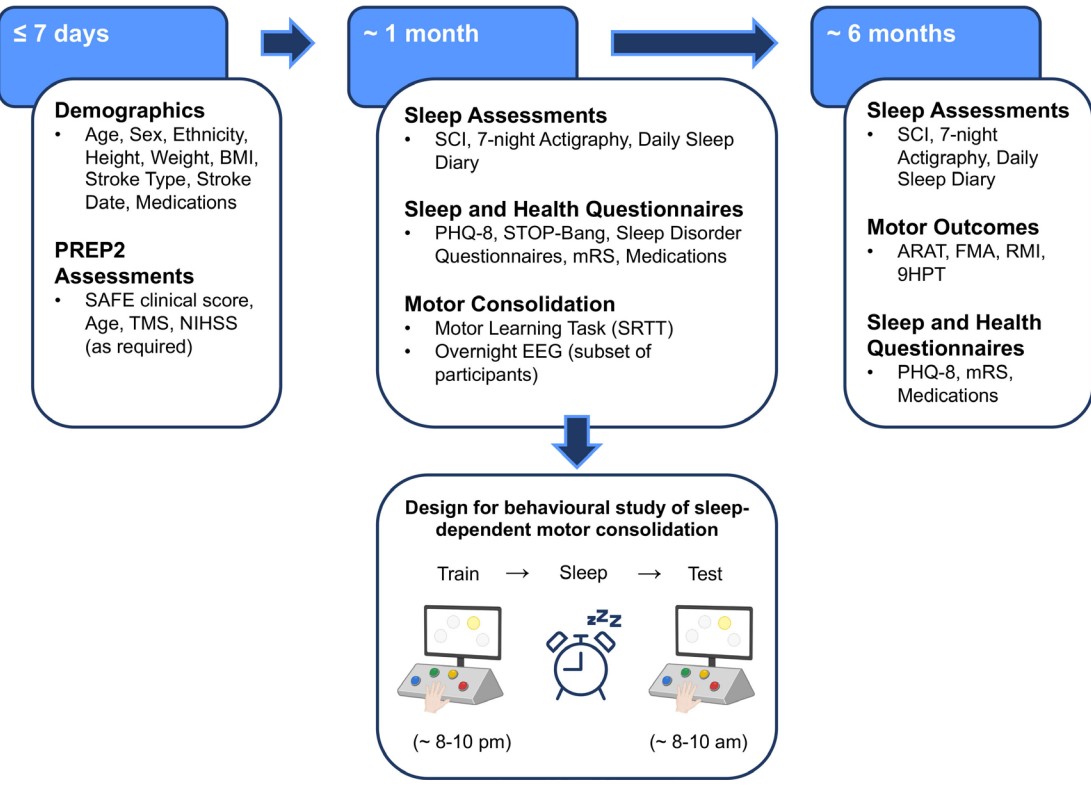

**Figure 2** Flowchart of the trial timeline and assessments. ARAT, Action Research Arm Test; BMI, body mass index; EEG, electroencephalography; FMA, Fugl-Meyer Assessment; mRS, modified Rankin Scale; NIHSS, National Institute of Health Stroke Scale; PHQ-8, Patient Health Questionnaire-8; RMI, Rivermead Mobility Index; SAFE, Shoulder Abduction Finger Extension; SCI, Sleep Condition Indicator; SRTT, Serial Reaction Time Task; STOP-Bang, Obstructive Sleep Apnoea Questionnaire; TMS, transcranial magnetic stimulation; 9HPT, Nine Hole Peg Test.

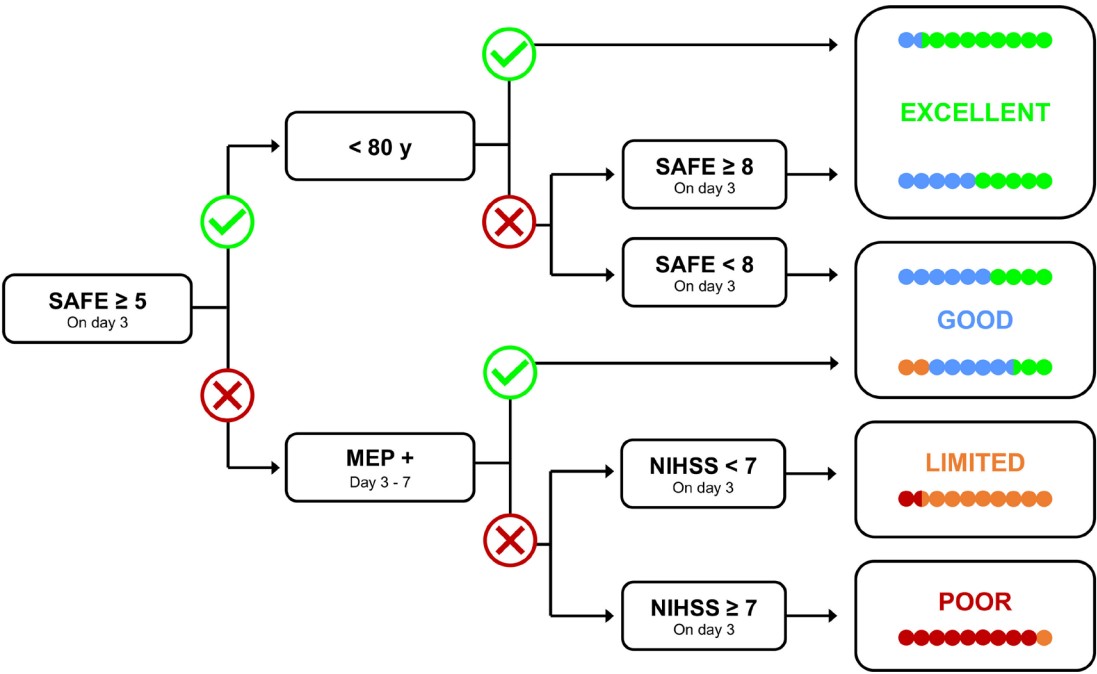

**Figure 3** Predict Recovery Potential algorithm (PREP2), included with permission, under license CC BY-NC-ND 4.0 (https://presto.auckland.ac.nz).[36] The algorithm uses neurological and clinical biomarkers to predict four different categories of potential recovery outcomes at 3 months after stroke. MEP, motor evoked potential; NIHSS, National Institute of Health Stroke Scale; SAFE, Shoulder Abduction Finger Extension Score.

the screening checklist—we will use surface electromyography to record muscle activity from the extensor carpi radialis and first dorsal interosseus of the affected upper limb between 3 and 7 days after stroke. A patient will be classified as MEP+ if MEPs of any amplitude are elicited with a consistent latency in response to at least five stimuli at rest or during voluntary muscle facilitation and MEP– if no MEPs can be elicited at 100% maximum stimulator output either at rest or while attempting voluntary muscle facilitation. We estimate that approximately 30% of participants will require TMS.[36] If a participant is unable to undergo TMS, we will use all other biomarker information available and categorise on a 'worst case scenario' basis (analyses will be conducted with and without these participants included to ensure that this estimation does not influence findings). Age and National Institute for Health Stroke Scale scores will also be collected to dictate final categorisation.

### 1 month post stroke

All procedures involved in the 1-month visit will be conducted at the location the patient is currently residing, with researchers visiting to collect data and set up equipment. At ~1 month after stroke (no more than 12 weeks), sleep measures will be collected via 7-night actigraphy recordings (measuring day/night activity, estimated total sleep time, WASO and sleep fragmentation index) and completion of the SCI.[26] For actigraphy measures, participants will wear a waterproof monitor on their least affected wrist (MotionWatch V.8, CamNTech). A sleep diary, detailing approximately what time participants fell asleep and woke up for each of the seven nights,

will also be completed (figure 2). We have previously, and continue to, successfully implement similar actigraphy and diary-based approaches in multiple research projects comprising of over 200 patients, proving to be safe and tolerable.[22 23 38 39] We will assess the presence of other sleep disorders via completion of a sleep screening and STOP-Bang questionnaires, and mood by means of the PHQ-8. Additionally, we will ask participants to complete a series of questions to determine the score for the mRS, and record current medications. We will record where the participant is residing at the time of the visit (eg, home, rehabilitation unit, care home, etc), and ask participants to self-report whether they have received motor rehabilitation and/or sleep therapy following their stroke, and if so, approximately how much (eg, sessions per week, average session length and upper and/or lower limb focus).

We will assess consolidation of motor sequence learning using an adapted version of the SRTT, see figure 2. The task has been adapted so that participants are required to move their entire upper limb and press each button with their whole hand rather than individual fingers as is customary in SRTTs,[12] see online supplemental file 1. Studies using similar tasks have reported effective learning in stroke patients using the paretic upper limb[40] and sleep-dependent motor improvements in older adult participants.[12] Participants will learn to press buttons on a button box with their affected limb following a repeated 8-item sequence. On-screen representations of each button will become highlighted, indicating which button to press. Once a response is given, the next target cue in

the sequence will become highlighted, and so on, regardless of whether the correct response is given. Participants will perform two different versions of the task (training and test), one in the evening and then one the following morning, respectively (figure 2). The training task will include 20 blocks of trials, with three repetitions of the 8-item sequence in each block. The test, aimed to determine sleep-dependent changes, will only include three blocks of trials to avoid confounds related to additional learning. To probe sequence specific learning, two blocks of 'random' sequences will be inserted at the beginning and towards the end of each of the training and test tasks. We will not explicitly inform participants of the sequence structure; however, a banner will appear at the top of the screen during every block, indicating whether the button presses will follow the sequence or a random order. At the end of the test task, participants will be asked to attempt to replicate as much of the sequence order as they can remember/were aware of without being shown the cues. Participants will be asked to train on the task between 20:00 and 22:00 or approximately 30–60 min before they intend to go to sleep and then be tested the following morning between 8:00 and 10:00 or at least 30 min after wake. The training will take approximately 20–30 min and retest will take no more than 10 min. Measures of both speed (response times) and accuracy will be recorded.

In a subset of participants, we will also measure overnight brain activity using a portable low-density EEG device. Evidence suggests that available products are efficacious for monitoring sleep-related electrophysiological signals and are comparable to gold-standard polysomnography protocols.[41 42] We have also previously explored the feasibility of portable at-home EEG device use by stroke survivors at home (unpublished data), with no major issues reported. Participants will be asked to wear the EEG device for three consecutive nights: one acclimatisation night followed by the night prior to completing the motor learning task, and the night of the motor learning task. Only participants who are residing at home will be invited for this assessment and they may opt out with no consequence to any of the other aspects of the study.

## 6 months post stroke

We will assess sleep using the same protocol as detailed for the 1-month visit (SCI, 7-night actigraphy and daily sleep diary). Similarly, we will measure mood (PHQ-8), disability (mRS) and record current medications (figure 2). We will also document where the participant is residing, and self-report of whether they have received motor rehabilitation and/or sleep therapy (and if so, how much) since the previous assessment.

To determine functional recovery 6 months after stroke, we will carry out a number of standardised clinical assessments requiring participants to make everyday movements with their hands, arms and legs. These include the ARAT, upper and lower extremity sections of the FMA, the RMI and the 9HPT. Again, researchers will visit participants in their homes or where they are currently residing to carry out all aspects of the 6-month assessment. In instances where participants prefer not to complete the movement assessments at their residing location, they will be invited to attend the research lab to complete these.

## Data management

Direct access to data will be granted to authorised representatives from the trial sponsor and host institution for monitoring and/or audit of the study in compliance with regulations. Participants will be assigned a unique study code, which will be used on all study documents (apart from consent forms) and all documents will be stored safely in confidential conditions. Anonymous data will be shared openly through a data sharing initiative (eg, Open Science Framework) following publication or public release of the study results.

## Data/statistical analyses

### Primary analyses

To test whether sleep in the subacute phase of recovery after stroke explains variability seen in upper limb motor outcomes, we will use the general linear model to test the association between sleep measures at 1 month and ARAT score at 6 months post stroke. We will include relevant covariates such as age, side affected, anticipated recovery outcome category (from PREP2 algorithm), PHQ-8 score, likely presence of sleep disorders and the activity metric from actigraphy recordings, etc as appropriate. To determine whether this relationship depends on motor consolidation, we will test for a mediation effect of motor consolidation at 1 month on this association. Motor consolidation will be determined by calculating the difference in performance from the end of evening training relative to the morning retest, with random trial blocks used to evaluate any sequence specific effects.

### Secondary analyses

To test whether motor consolidation mediates the relationship between sleep and a broader range of clinical outcomes and timepoints, we will use the general linear model, testing for a mediation effect of motor consolidation at 1 month post stroke on the correlation between sleep measures at 1 month and ARAT, FMA, 9HPT and RMI scores at 6 months post stroke.

### Exploratory analyses

To explore whether motor consolidation is associated with specific sleep signals after stroke, we will use the general linear model to correlate motor consolidation with EEG derived sleep metrics at 1 month post stroke. Finally, to explore changes in sleep alongside recovery, we will use the general linear model to test for changes in sleep measures from 1 month to 6 months post stroke.

## Study monitoring

The study management group, consisting of the principal investigator and members of the research team, will meet multiple times per year to monitor the progress of the study. Participants will be regularly monitored, and data

evaluated to check for protocol compliance and accuracy in relation to source documents. Responsible personnel from the NHS Trust(s) and/or sponsor may also monitor or audit the trial where appropriate.

Throughout the trial timeline, we will check with participants for the occurrence of any adverse events (AEs). Potential expected minor AEs include distress associated with completing mood questionnaires and skin irritation from wearing the actigraphy monitor. Any AEs that do occur during study enrolment will be documented and participants will be advised on the most appropriate course of action, for example, withdrawing from particular assessments or contacting their GP. There is a potential, but very small, risk of seizure with TMS. The most recent update to safety guidelines suggested that, given the large number of patients who have received TMS and the small number of seizures, the risk of TMS to induce seizures is very low.[43] All participants who require TMS will be screened prior to testing and all procedures will be conducted within the safety guidelines. There are no anticipated serious AEs (SAEs) associated with the present trial. Any SAEs that occur (and deemed unexpected and related to research procedures) will be reported to the appropriate ethics committee within 15 days of awareness of the event, using the Health Research Authority (HRA) report of SAE form (see HRA website for further info).

During the course of the trial, a participant may choose to withdraw at any timepoint and for any reason. The reason for withdrawal will be recorded if known, but participants are not obliged to provide a reason. Data from withdrawn participants may be kept and used for analyses as appropriate. If there is sufficient time to conduct all necessary study assessments within the study period, additional participants will be recruited to replace withdrawals. If a participant is unable to comply with one aspect of the assessment (eg, the motor learning task), then they may continue other assessments unless they choose to withdraw entirely. Moreover, an investigator may choose to discontinue a participant from the trial if it is considered necessary for any reason, for example, significant protocol deviation, non-compliance, ineligibility (either arising during the study period or retrospectively having been overlooked at screening) etc.

### Patient and public involvement

Stroke survivors were involved in producing and reviewing all participant facing documents. We also encouraged patients to give feedback on the motor learning task at different public/patient engagement events throughout the design of the study and during pilot testing. This feedback was used to write instructions and edit task difficulty. We will seek further assistance from patient/public contributors when disseminating results from the trial—in the form of lay summaries—to ensure key messages are communicated in a meaningful way to stroke communities.

### ETHICS AND DISSEMINATION

This trial has received both HRA, Health and Care Research Wales and National Research Ethics Service approval (IRAS: 304135; REC: 22/LO/0353). The study is registered on a publicly accessible database (clinicaltrials.gov: NCT05746260). The results of the study will be uploaded to clinicaltrials.gov within 12 months of the end of the trial and disseminated via presentations at scientific conferences, peer-reviewed publication, public engagement events, stakeholder organisations and other forms of media where appropriate.

### Limitations

Despite efforts to develop a robust study design, there remain some limitations to the present protocol. First, due to logistical constraints, consolidation of motor learning is only assessed at one timepoint, approximately 1 month post stroke. We are working on the assumption that consolidation on any given night will be determined by a combination of *stable factors* (such as habitual sleep quality) as well as factors *specific to that night* (such as whether someone's sleep was disturbed by external noises). Our study is designed to capture the former and not the latter. To capture the former, measuring consolidation on any night can provide an estimate of habitual consolidation performance. To capture the latter would require assessments of consolidation on multiple nights following rehabilitation sessions, which would be too complex to administer and too burdensome for participants. Second, the accuracy of sleep measurements may be limited due to being derived mainly from actigraphy recordings (eg, movement related), as well as EEG recordings collected at only one timepoint post stroke for a subset of participants. Despite this, actigraphy-derived sleep measures have been shown to relate to motor outcomes over periods of rehabilitation.[23] Finally, the self-report nature of some key outcomes, for example, SCI may limit outcome sensitivity.

### Summary and conclusions

The present trial aims to investigate the relationship between sleep, motor consolidation and clinical outcomes after stroke. The importance of motor learning and neuroplasticity in poststroke recovery is widely recognised,[4 8 44] however, the role of sleep in offline consolidation of motor memories is an area that has yet to be fully explored in the context of stroke rehabilitation. We will investigate whether sleep measures in the subacute stage of stroke explain variability in recovery of upper limb function, assessing the relationship between sleep and recovery in terms of motor consolidation and determining whether motor consolidation mediates the relationship between sleep and clinical outcomes. Using wearable remote technologies, we will also explore electrophysiological markers of sleep after stroke, changes in sleep alongside recovery and correlations between sleep and motor function. We aim to build on our previous work showing that sleep is disrupted after stroke/brain injury

and predicts poorer recovery,[22 23] to determine whether the consolidation of motor memories is a candidate process to mediate this relationship. We hope the findings of this study will help to enhance our understanding of the role of sleep in stroke recovery and may inform the development of novel interventions for improving post-stroke rehabilitation outcomes.

## Rights retention

This research is funded in whole, or in part, by the Wellcome Trust (grant number 222446/Z/21/Z). For the purpose of open access, the author has applied a CC BY public copyright license to any author accepted manuscript version arising from this submission.

#### Author affiliations

[1] Wellcome Centre for Integrative Neuroimaging (WIN), Nuffield Department of Clinical Neurosciences, University of Oxford, Oxford, UK
[2] Buckinghamshire Oxfordshire and Berkshire West Integrated Care Board (BOB ICB), Oxford, UK
[3] MRC Stroke Unit, Oxford Centre for Enablement, Oxford University Hospitals NHS Foundation Trust, Oxford, UK
[4] Wolfson Centre for the Prevention of Stroke and Dementia, Nuffield Department of Clinical Neurosciences, Oxford University, Oxford, UK

**Acknowledgements** We would like to thank the individuals at Oxfordshire Stroke Rehabilitation Unit (OSRU) andNational Institute for Health and Care Research Clinical Research Network (NIHR CRN) Stroke for their support in setting up, recruiting for, and facilitating the trial. Thanks also to Sebastian Rieger for his input regarding behavioural motor task hardware design.

**Contributors** MW, BR, MKF and HJ-B conceived and designed the protocol with input from MPM. EG, RT and ND facilitate recruitment, consent process and data collection. AR-B designed and constructed the hardware for the behavioural motor task. MW and BR were involved in the day-to-day running of the trial. MW drafted the manuscript. All authors edited, revised and approved the final version of the manuscript.

**Funding** This study is funded by the Wellcome Trust (Principal Research Fellowship to HJ-B; 222446/Z/21/Z) and supported by the National Institute for Health and Care Research (NIHR) Oxford (Oxford University Hospitals) and Oxford Health Biomedical Research Centres. These funding sources had no role in the design of this study and will not have any role during its execution, analyses or interpretation of data. The Wellcome Centre for Integrative Neuroimaging is supported by core funding from the Wellcome Trust (203139/Z/16/Z and 203139/A/16/Z).

**Competing interests** None declared.

**Patient and public involvement** Patients and/or the public were involved in the design, or conduct, or reporting, or dissemination plans of this research. Refer to the Methods and analysis section for further details.

**Patient consent for publication** Not applicable.

**Provenance and peer review** Not commissioned; externally peer reviewed.

**ORCID iDs**
Matthew Weightman http://orcid.org/0000-0003-4379-2725
Barbara Robinson http://orcid.org/0000-0002-1721-7682
Melanie K Fleming http://orcid.org/0000-0003-2232-9598

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
