## [Reviewer comments · BMJ Open]

ARTICLE DETAILS

TITLE (PROVISIONAL)	Sleep and motor learning in stroke (SMiLES) - a longitudinal study investigating sleep-dependent consolidation of motor sequence learning in the context of recovery after stroke
AUTHORS	Weightman, Matthew; Robinson, Barbara; Mitchell, Morgan; Garratt, Emma; Teal, Rachel; Rudgewick-Brown, Andrew; Demeyere, Nele; Fleming, Melanie K.; Johansen-Berg, Heidi

VERSION 1 – REVIEW

REVIEWER	Kelly Westlake University of Maryland Baltimore
REVIEW RETURNED	08-Sep-2023

GENERAL COMMENTS	Comments to Author The authors have proposed a study to determine whether sleep in the sub-acute phase of stroke recovery explains the variability in upper limb motor outcomes after stroke and whether this relationship is dependent on consolidation of motor learning. This is a compelling protocol, which has the potential to enhance clinical practice. The authors clearly identify gaps in the current literature, justifying their research questions. Primary, secondary, and exploratory research aims are clearly stated. Additionally, the study inclusion criteria, exclusion criteria, and outcome measures appear mostly appropriate for the study questions and design. Strengths of this research include the involvement of stroke survivors in the production and review of all participant-facing documents and motor tasks. Minor revisions are being requested for purposes of clarity and rigor as outlined below. Minor considerations. 1. If statistics are used, are they appropriate and described fully? Partially Use of general linear modeling is planned for primary, secondary, and exploratory analysis. For primary analysis, anticipated recovery time and PHQ-8 will be used as covariates. It is recommended that the authors consider use of additional covariates controlling for age, dominant/non-dominant upper extremity involvement, and time spent in rehabilitation in linear modeling. Additionally, please state how consolidation will be calculated. It is currently unclear whether consolidation will be treated as the difference between all or a subset of PM versus AM trials. 2. Are the study limitations discussed adequately? Partially The authors list major limitations including (1) heterogeneous subject settings and (2) use of recruiting techniques that may limit
---

	sample diversity. The following additional limitations should be included: (1) use of actigraphy (potential for limited accuracy due to partial reliance on subject movements to indicate sleep/wake state) and self-report (potential for limited accuracy due to possible cognitive deficits post-stroke) without objective EEG recording to index sleep quality and duration at 6 month timepoint and (2) between-subject variability in the amount of upper extremity therapy/treatment. Also, evaluation of consolidation at a single time point five months prior to assessing motor performance may be a limitation in research design. Subject sleep patterns may change over this five-month period (related to e.g., recovery stage or setting) potentially influencing processes of consolidation. It is recommended that the authors consider including the lack of consolidation assessment at multiple timepoints as a limitation. 3. Research aims: Please provide greater detail on e.g., stage of stroke recovery, recovery in reference to more versus less affected upper extremity, and subject population be added to the primary research aims for clarity. 4. Inclusion and exclusion criteria: Please state whether dominant arm affected and/or non-dominant arm affected stroke survivors will be included in this sample. Additionally, the authors should consider the exclusion of adults with known sleep disorders prior to stroke event who may have pre-existing impairments consolidation processes or justify reasons why they are not excluded. 5. Text inconsistencies. Please revise or clarify the minor inconsistency in reporting of subject recruitment/assessment window. In the inclusion criteria, the recruitment window is noted as ≤ 7 days post-stroke; however, on page 4 line 74 (see “Strengths and Limitations”) the early assessment window is noted as < 7 days post-stroke.
--	--

REVIEWER	Kara Patterson University of Toronto, Physical Therapy
REVIEW RETURNED	15-Sep-2023

GENERAL COMMENTS	BMJOPEN-2023-077442 Thank you for the opportunity to review this interesting manuscript. The authors outline the protocol for a longitudinal study that will follow approximately 100 people with stroke for 6 months, measuring various markers of sleep, motor consolidation, and upper limb recovery and motor control. The primary aim of the study is to understand if measures of motor consolidation mediate the relationship between sleep and motor outcomes in people with stroke. There are also several secondary objectives. Stroke is a significant international health concern. Given the relationship between motor learning and stroke rehabilitation, the results of this study have the potential to make a significant impact on clinical practice. The protocol is clear and well-written. I have a few concerns that are outlined below. 1) Pg 5 In 85. The authors note that long-term motor impairment after stroke is affected “especially of the upper limb”. It would help the reader if the authors elaborated why this is important
--

	to provide context for the potential impact of the study. 2) Pg 6, ln 108. The authors note sleep disruption is common after stroke. Again, for the general reader, it would be helpful to provide a few examples of what is meant by sleep disruption which the reviewer assumes is an umbrella term. 3) Pg 8, ln 169-172. The authors provide the assumptions made, but details about the sample size calculation are absent. If applicable, can the author provide more details about how sample size was calculated? 4) The proposed relationship between sleep, consolidation and motor recovery could be more clearly defined. It could be made clearer why consolidation only measured at one timepoint at 1 month post stroke could explain motor recovery at 6 months. Is it not important to know the consolidation that occurs after rehabilitation sessions that the participant receives over those 6 months? Is it reasonable to assume that consolidation at 1 month represents the consolidation that may occur at several points during rehab over the subsequent 6 months? 5) Related to point 3, the reader would benefit from a figure that illustrates the proposed relationships and connections between all the factors and concepts raised by the authors (i.e., sleep quality, consolidation, motor recovery, sleep disturbances, predicted recovery). 6) Who will be completing the sleep diary? The participant, caregiver, researcher? Are there procedures in place to ensure they are completed and if they are accurate? 7) The reviewer questions whether a retrospective report on how much rehabilitation was received will be accurate or meaningful. Further to this point, will the volume and intensity of the therapy received be considered? These are two training factors that are strongly related to motor learning and motor outcomes post stroke. Would these have potential to mediate the relationship between consolidation and motor outcomes?
--	---

VERSION 1 – AUTHOR RESPONSE

REVIEWER #1

1) If statistics are used, are they appropriate and described fully? Partially

Use of general linear modeling is planned for primary, secondary, and exploratory analysis. For primary analysis, anticipated recovery time and PHQ-8 will be used as covariates. It is recommended that the authors consider use of additional covariates controlling for age, dominant/non-dominant upper extremity involvement, and time spent in rehabilitation in linear modeling. Additionally, please state how consolidation will be calculated. It is currently unclear whether consolidation will be treated as the difference between all or a subset of PM versus AM trials.

Apologies for the lack of clarity, we didn't intend to suggest that those would be the only covariates but rather examples of relevant covariates we anticipate needing to include. We have added other covariates to be considered as well as "etc" to try to make this clearer.

Line 399-401: "We will include relevant covariates such as age, side affected, anticipated recovery outcome category (from PREP2 algorithm), PHQ-8 score, likely presence of sleep disorders, and the activity metric from actigraphy recordings, etc. as appropriate."

In reference to consolidation, yes, we will compare a subset of the PM trials (e.g., the end of learning) to compare to the retest trials, with the random blocks used to identify any sequence specific effects. We imagine this will be approximately trials within the last 4 or 5 blocks. However, as there is limited

existing data in this cohort, specifically using the paretic upper limb, we are unsure when learning is likely to plateau during the PM session and therefore are reluctant to make a hard decision without understanding the behaviour more.

We have tried to make this analysis plan for the consolidation clearer in the corrected manuscript.

Line 404-406: "Motor consolidation will be determined by calculating the difference in performance from the end of evening training relative to the morning retest, with random trial blocks used to evaluate any sequence specific effects."

2) Are the study limitations discussed adequately? Partially

The authors list major limitations including (1) heterogeneous subject settings and (2) use of recruiting techniques that may limit sample diversity. The following additional limitations should be included: (1) use of actigraphy (potential for limited accuracy due to partial reliance on subject movements to indicate sleep/wake state) and self-report (potential for limited accuracy due to possible cognitive deficits post-stroke) without objective EEG recording to index sleep quality and duration at 6 month timepoint and (2) between-subject variability in the amount of upper extremity therapy/treatment. Also, evaluation of consolidation at a single time point five months prior to assessing motor performance may be a limitation in research design. Subject sleep patterns may change over this five-month period (related to e.g., recovery stage or setting) potentially influencing processes of consolidation. It is recommended that the authors consider including the lack of consolidation assessment at multiple timepoints as a limitation.

Thank you highlighting this omitted information. The limitations already listed in the manuscript form part of the 'Strengths and Limitations' section (a section required for submission of a protocol paper to BMJ Open and "should be no more than 5 bullet points relating specifically to the methods - not the results of the study"). Therefore, we cannot add all the suggested limitations to this section. Instead, they have been added to the manuscript in an additional section.

Line 82-83: "Likely between-subject variability in the amount of upper extremity therapy/treatment received during study enrolment."

Line 478-494: "Despite efforts to develop a robust study design, there remain some limitations to the present protocol. Firstly, due to logistical constraints, consolidation of motor learning is only assessed at one timepoint, approximately 1-month post-stroke. We are working on the assumption that consolidation on any given night will be determined by a combination of stable factors (such as habitual sleep quality) as well as factors specific to that night (such as whether someone's sleep was disturbed by external noises on). Our study is designed to capture the former and not the latter. To capture the former, measuring consolidation on any night can provide an estimate of habitual consolidation performance. To capture the latter would require us to assess consolidation on multiple nights following rehabilitation sessions, which would be too complex to administer and too burdensome for participants. Secondly, the accuracy of sleep measurements may be limited due to being derived mainly from actigraphy recordings (e.g., movement related), as well as EEG recordings collected at only one timepoint post-stroke for a subset of participants. Despite this, actigraphy-derived sleep measures have been shown to relate to motor outcomes over periods of rehabilitation.²³ Finally, the self-report nature of some key outcomes e.g., SCI may limit outcome sensitivity."

Please also see Reviewer #2, comment 4 response for further clarification on the matter of a singular measurement of motor consolidation.

3) Research aims: Please provide greater detail on e.g., stage of stroke recovery, recovery in reference to more versus less affected upper extremity, and subject population be added to the primary research aims for clarity.

We are not entirely sure what is being requested by the reviewer, but we added more specific information to the 'Trial Objectives' section.

Hopefully, this is what is being requested.

Primary Objectives

1) Test whether sleep measures at ~1-month post-stroke explain variability in recovery of movement of the paretic upper limb at ~6 months.

2) Assess whether the relationship between sleep and recovery after stroke depends on motor consolidation.

Secondary Objective

1) Determine if motor consolidation at ~1-month post-stroke mediates the relationship between sleep and clinical outcomes, such as whole-body motor impairment, mobility, and hand function of the more-affected side, measured at ~6 months post-stroke.

Exploratory Objectives

1) Explore whether motor consolidation is associated with specific electrophysiological markers of sleep after stroke.

2) Investigate changes in sleep alongside recovery over the first 6 months after stroke.

3) Explore correlations between sleep disruption and motor function between ~1- and ~6-months post-stroke.

4) Inclusion and exclusion criteria: Please state whether dominant arm affected and/or non-dominant arm affected stroke survivors will be included in this sample. Additionally, the authors should consider the exclusion of adults with known sleep disorders prior to stroke event who may have pre-existing impairments consolidation processes or justify reasons why they are not excluded.

There is no specific requirement for inclusion based on the side affected by the stroke with regards to limb dominance, but we will record this information as part of the participant characteristics data. This info has been added to the inclusion/exclusion criteria.

In respect to pre-existing sleep disorders, we don't exclude participants on these grounds. We are primarily interested in sleep between the various study timepoints and the potential impact of sleep disruption on recovery, regardless of the presentation of sleep/sleep disorders prior to stroke (which will vary greatly between participants). Further to this, there may be cases in which individuals have undiagnosed sleep disorders pre-stroke, who would be difficult to identify retrospectively and exclude on that same basis. However, we do screen patients at 1-month for a variety of different potential sleep disorders, which can be used as covariates in any analysis where appropriate.

5) Text inconsistencies. Please revise or clarify the minor inconsistency in reporting of subject recruitment/assessment window. In the inclusion criteria, the recruitment window is noted as ≤ 7 days post-stroke; however, on page 4 line 74 (see "Strengths and Limitations") the early assessment window is noted as < 7 days post-stroke.

Thank you for highlighting this inconsistency. We have corrected accordingly – the window is on or within 7-days and thus (\leq), please see below.

Line 74: "Longitudinal study, with early assessment (≤ 7 days post-stroke) and follow-ups throughout sub-acute recovery."

REVIEWER #2

1) Pg 5 In 85. The authors note that long-term motor impairment after stroke is affected “especially of the upper limb”. It would help the reader if the authors elaborated why this is important to provide context for the potential impact of the study.

After consideration, this sentence does not necessarily add anything to text and thus has been removed to avoid vagueness.

2) Pg 6, In 108. The authors note sleep disruption is common after stroke. Again, for the general reader, it would be helpful to provide a few examples of what is meant by sleep disruption which the reviewer assumes is an umbrella term.

Thank you for this comment, we have now expanded this sentence to give some examples of how sleep is disrupted after stroke.

Line 110-112: “Notwithstanding, sleep disruption is commonplace after stroke, with more fragmented sleep, reduced sleep efficiency, and less total sleep time commonly reported (19-22).”

3) Pg 8, In 169-172. The authors provide the assumptions made, but details about the sample size calculation are absent. If applicable, can the author provide more details about how sample size was calculated?

Apologies for not including more information regarding the sample size calculations. This has now been updated in the revised manuscript.

Lines 175-179: “Based on findings from previous work in inpatients with brain injury (23), we predict that sleep measures increase variance in motor outcomes explained by ~10%. Therefore, to achieve a power of 0.9 (1- β error probability) with a significance level (alpha) of < 0.05, a total sample size of $\sim n = 100$ is recommended (calculated using G*Power; linear multiple regression, with a fixed model). In anticipation of participant attrition, we will aim to recruit 150 participants.”

4) The proposed relationship between sleep, consolidation and motor recovery could be more clearly defined. It could be made clearer why consolidation only measured at one timepoint at 1 month post stroke could explain motor recovery at 6 months. Is it not important to know the consolidation that occurs after rehabilitation sessions that the participant receives over those 6 months? Is it reasonable to assume that consolidation at 1 month represents the consolidation that may occur at several points during rehab over the subsequent 6 months?

We are working on the assumption that consolidation on any given night will be determined by a combination of stable factors (such as habitual sleep quality) as well as factors specific to that night (such as whether someone’s sleep was disturbed by external noises on that particular night). Our study is designed to capture the former and not the latter. To capture the former, measuring consolidation on any night can provide an estimate of habitual consolidation performance. To capture the latter would require us to assess consolidation on multiple nights following rehabilitation sessions, which would be too complex to administer and too burdensome for participants.

For our single timepoint of consolidation assessment, we chose ~ 1-month post stroke as we assumed patients may be more settled after their stroke and be in a more suitable position to carry out the various measurements and that their consolidation performance might be more stable. This timepoint can be relatively flexible depending on the individual patient’s situation. For example, if the participant is in the middle of being discharged from an inpatient rehabilitation unit at the 1-month

mark, it would be more suitable to delay the visit a week or two, so that they can readjust to the home environment (without further disruption associated with a study visit). This flexibility is necessary to limit participant attrition.

We hypothesise that inter-individual variation in consolidation, due to habitual and stable factors, explains some variation in recovery outcomes.

An interesting question for future work could be to assess whether motor consolidation performance (and variation in that across individuals) varies significantly over time. A related question for future work would be to address whether motor consolidation assessed on specific nights following rehabilitation sessions provides a better predictor of recovery outcomes than assessing consolidation at a single timepoint.

We have also included the single measurement of motor consolidation in a limitations section of the manuscript (see Reviewer #1, comment 2 response).

5) Related to point 3, the reader would benefit from a figure that illustrates the proposed relationships and connections between all the factors and concepts raised by the authors (i.e., sleep quality, consolidation, motor recovery, sleep disturbances, predicted recovery).

We thank the reviewers for this suggestion. However, as some of these relationships are speculative, we don't feel an additional schematic figure would be all that useful. We think it would be a good addition to the final study manuscript and have made a note to include, if appropriate.

6) Who will be completing the sleep diary? The participant, caregiver, researcher? Are there procedures in place to ensure they are completed and if they are accurate?

Overall, we expect the circumstances to differ for each participant. Some participants may feel comfortable and capable to fill out the diary on their own. Others may require the help of a spouse, family member, or carer. Finally, inpatients may require the help of their clinical teams. At each one-month visit, we will carefully describe all study components. We will include family/carers/clinical teams in these discussions as appropriate to make sure the diaries are completed as accurately as possible. It is important to note that the sleep diary is just to help guide the actigraphy analysis and is not an outcome itself.

7) The reviewer questions whether a retrospective report on how much rehabilitation was received will be accurate or meaningful. Further to this point, will the volume and intensity of the therapy received be considered? These are two training factors that are strongly related to motor learning and motor outcomes post stroke. Would these have potential to mediate the relationship between consolidation and motor outcomes?

We will record data regarding the number of sessions per week, the average time of each session, and the focus of the rehab (upper limb, lower limb, both etc). This information has now been included in the manuscript:

Lines 324-327: "... to self-report whether they have received motor rehabilitation and/or sleep therapy following their stroke, and if so, approximately how much (e.g., sessions per week, average session length, and upper and/or lower limb focus)."

We understand this retrospective self-report may not be hugely accurate, but it will allow us to approximate the variability in rehab prescribed across participants and therefore can be used as a covariate to analysis if necessary.

VERSION 2 – REVIEW

REVIEWER	Kara Patterson University of Toronto, Physical Therapy
REVIEW RETURNED	08-Jan-2024
GENERAL COMMENTS	Thank you for the opportunity to review this revised manuscript of a study protocol. I appreciate the authors thoughtful and thorough responses to the concerns I raised. The authors have fully addressed all concerns and I have no further comments.